

# Spatial-temporal variability of *Mytilus galloprovincialis* Lamarck 1819 populations and their accumulated sediment in northern Portugal

Catarina Ramos-Oliveira[1,2], Leandro Sampaio[1,2], Marcos Rubal[1,2] and Puri Veiga[1,2]

[1] CIIMAR Interdisciplinary Centre of Marine and Environmental Research, Matosinhos, Portugal
[2] Deparment of Biology, University of Porto, Porto, Portugal

Corresponding author
Marcos Rubal, marcos.garcia@fc.up.pt

## ABSTRACT

*Mytilus galloprovincialis* is an ecosystem engineer that provides habitat and generates environmental heterogeneity, increasing local biodiversity. Moreover, it is an economically important species representing 14% of the global production of marine bivalves. Natural drivers and the increase of anthropogenic pressures, such as sediment stress, influence its populations on rocky shores. The objective of this study was to explore the spatial-temporal patterns of different *M. galloprovincialis* attributes along the north of Portugal. For that purpose, six rocky shores were selected and sampled six times along the year 2019. The percentage of cover, density, spat density, condition index, clump thickness, size classes and clump sediment content were considered. Results showed the lack of seasonality in *M. galloprovincialis* along the north coast of Portugal. However, density, spat stage, clump thickness, condition index and size classes showed some variability among dates and shores. The percentage cover and sediment content only significantly differed among shores. Our results indicated an absence of seasonality for all the studied variables, probably because temperature was always within the optimum range for this species and the abundance of food supply in the study area independently of the season. The accumulated sediment on mussel clumps did not show any temporal variability with only significant differences among shores. The accumulated sediment was composed mainly by medium and coarse sand and it was correlated with mussel average size, condition index, but especially with the mussel clump thickness.

## INTRODUCTION

Mussels are important organisms in intertidal systems because they keep biodiversity, sheltering, supporting and enhancing diverse assemblages of invertebrates (*Leigh Jr et al., 1987*; *Seed & Suchanek, 1992*; *Seed, 1996*) inducing physical changes in the substrate and providing suitable space and resources for different species of animals and macroalgae (*Thiel & Ullrich, 2002*; *Loehle, 2004*; *Prado & Castilla, 2006*; *Borthagaray & Carranza, 2007*). The Mediterranean mussel *Mytilus galloprovincialis* Lamark, 1819 is one of the most frequent and abundant native mussel species in the south Atlantic European and Mediterranean

Sea rocky shores (*Braby & Somero, 2006*). Recently, *Lynch et al. (2020)*, have detected a northward range expansion of *M. galloprocincialis* along Irish shores, probably as a result of global warming. Additionally, *M. galloprovincialis* is also a widespread invasive species in other regions like South Africa (*De Moor & Bruton, 1988*). Moreover, *M. galloprovincialis* is an economically important species because of its high production (*Guo et al., 2018*). Because of that, it is vulnerable to recreational and commercial harvesting (*Smith & Murray, 2005*). Aquaculture of marine bivalves accounts 14% of the global marine production (*Wijsman et al., 2019*). However, for mussels this practice, requires the exploitation of wild populations, involving the collection of young mussels' larvae directly from the water or harvesting of small mussels (0.5–2 cm) from intertidal and subtidal beds (*Consellería de Pesca, Marisqueo y Acuicultura, 2000*; *Cáceres-Martínez & Figueras, 2007*; *Figueras, 2007*). This practice affects not only mussel juveniles but also adult stocks because it slows down the recovery of mussels (*Harris et al., 2003*), and thus it may influence the whole communities associated with mussels (*Veiga et al., 2020*).

Natural populations of *M. galloprovincialis*, are under the effects of natural and anthropogenic disturbances (*Nicastro, McQuaid & Zardi, 2019*), which can have a strong influence on natural population dynamics (*UNE, 2006*). Intertidal populations of *M. galloprovincialis* often live in mechanically stressful environments, being affected by natural ecological drivers like changes of temperature, salinity, concentration of suspended matter or phytoplankton (*Braby & Somero, 2006*; *Zardi et al., 2008*; *Garner & Litvaitis, 2013*; *Araújo et al., 2020*) and the presence of filamentous algae that promote the larval colonization (*Ceccherelli & Rossi, 1984*). Among anthropogenic disturbances, coastal urbanisation is one of the most persistent and increasing threats due to an increase of ocean sprawl and high diversity of pollutants coming from industry and domestic products (*Todd et al., 2019*). Coastal urbanisation has proved to have negative effects on the abundance and size structure of intertidal *M. galloprovincialis* (*Veiga et al., 2020*). Other anthropogenic disturbances like harvesting or trampling have also proved to have negative effects on mussels populations (*Smith & Murray, 2005*). Among the multiple stressors that can affect mussel populations, the sediment accumulated on rocky shores has been poorly studied. *Airoldi (2003)* suggests that sediment can interfere in the development of mussel beds, particularly the accumulation of sediment interferes with physical and biological processes that lead to a reduction of suitable habitats (*Airoldi, 2003*). The main negative effect described for sediment accumulation on mussel populations is the burial (e.g., *Hutchison et al., 2016*; *Dos Santos et al., 2018*) but that effect can be different among mussel species in function of their ability to emerge or resist burial (*Zardi et al., 2008*; *Hutchison et al., 2016*; *Hutchison et al., 2020*). Moreover, the sediment can be a reservoir of contaminants that can result toxic for marine animals (*Long et al., 1996*; *Fichet, Radenac & Miramand, 1998*). Finally, sediment accumulation can also modify the services provided by mussels modifying the invertebrate assemblage diversity associated to mussel beds and facilitating the colonisation of mussel beds by soft bottoms macro- and meiobenthic species (e.g., *Dos Santos et al., 2018*).

Studies about patterns of variation in time and space of natural populations and assemblages are a chief goal in ecology, especially when they include economically

relevant species with a key ecological role like *M. galloprovincialis* (*Levin, 1992*; *Underwood, Chapman & Connell, 2000*). The understanding of variation in populations is essential to the management and conservation of species because, as natural and anthropogenic processes like sediment accumulation may be very variable in space and time, their effect on the population dynamics will also be very variable (*Andrew & Mapstone, 2006*; *Levin, 1992*; *Underwood & Chapman, 1996*; *Benedetti-Cecchi, 2000*; *Bertocci et al., 2012*). Bivalve molluscs show high plasticity on their responses and adaptation to local conditions (*Bayne & Widdows, 1978*; *Widdows et al., 1984*; *Tedengren et al., 1990*; *Navarro et al., 1991*) so their settlement and growth varies among regions, according to the local environment, but it can also vary along the year in response to oceanographic natural oscillations (*Seed, 1976*). In sedentary animals like mussels, their reproduction and growth are the only mechanisms to sustain populations (*Seed, 1976*). Consequently, exogenous and endogenous factors are crucial for controlling physiological rates and the reproduction (*Orban et al., 2002*; *Nagarajan, Lea & Goss-Custard, 2006*; *Seed, 1976*). Therefore, knowledge of the natural variability in time and space is decisive to understand population dynamics (*Underwood, 1981*) and for the management and conservation of the stocks of commercially exploited species, as well as the services provided by those species.

Nowadays, there is a lack of information about the spatial–temporal variability of *M. galloprovincialis* populations along the Atlantic coast of the Iberian Peninsula, where this species plays key ecological and economical roles. Similarly, the ability of *M. galloprovincialis* clumps to accumulate sediment on rocky shores is almost unexplored. The main aims of this study are to fill these gaps of knowledge and to explore the spatial–temporal patterns of *M. galloprovincialis,* and accumulated sediments on their clumps, along rocky shores in the north of Portugal. Different variables will be considered: density, spat stage, percentage cover, clump thickness, condition index and different size classes of mussels to achieve these objectives. Moreover, the spatial–temporal patterns of sediment accumulation in mussel clumps and its relationships with the mussel population attributes will be examined.

## MATERIALS & METHODS

### Study area

The study was done between January 2019 and December 2019 and included six rocky shores along 90 km of the northern Portuguese coast: Aguda (N 41°04246′, W 8°653254′) it is an urban schist platform shore with significant natural/anthropogenic sediment input and close to an artificial seawall, Valadares (N 41°089167′, W 8°658374′) it is an urban schist platform shore with significant natural/anthropogenic sediment input, Cabo do Mundo (N 41°225741′, W 8°717976′) it is an urban granitic shore with gentle slope in a very industrial area, Carreço (N 41°742040′, W 8°878418′) it is a non-urban schist shore with gentle slope and significant natural sediment input, Forte do Cão (N 41°798244′, W 8°874848′) it is a non-urban granitic shore with gentle slope close to a storm-water outflow and Moledo (N 41°822605′, W 8°874894′) it is a non-urban granitic shore with gentle slope (Fig. 1). Along the northern Portuguese coast, intertidal area varies from soft to hard bottoms but they mostly present a mixture of both substrates. Sampling was done during low tide in the mid

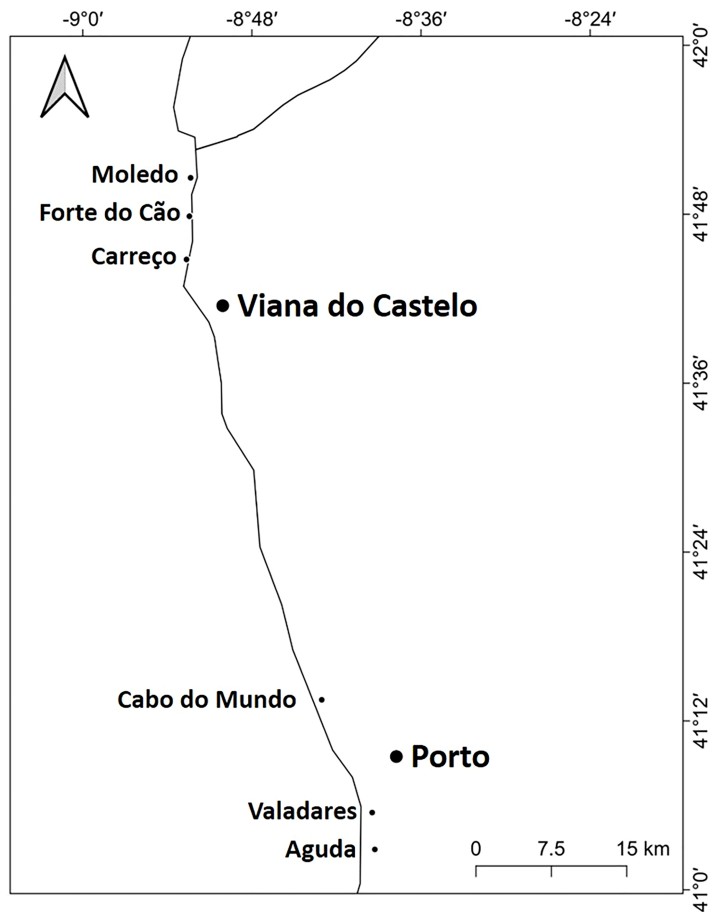

**Figure 1 Studied area.** Location of the 6 studied shores along the Atlantic coast of Northern Portugal.

tide level of the rocky shores, that typically is dominated by the species *M. galloprovincialis* (*Veiga et al., 2020*). In the north of Portugal, the spring–summer and the autumn–winter seasons are characterized by strong differences in mean monthly precipitation, air and water temperature, hydrodynamic conditions, wave height and storm frequency (*Dias et al., 2002*; *Bertocci et al., 2012*). The coastline is largely straight and is exposed to wave action, which varies strongly with seasons (*Dias et al., 2002*). In the spring–summer period the typical wave heights are between 1 and 3 m, with periods of 11–13 s (*Dias et al., 2002*) and swells direction generally is from W and NW. In the autumn–winter period, the waves can reach around 7 m with periods of 13–18 s (*Dias et al., 2002*). The tidal regime of the study area is semidiurnal with the largest spring tides of 3.5–4.0 m. Moreover, the studied area is subjected to a seasonal upwelling that provides nutrients and increases the primary production in the water column during spring and summer months (*Lemos & Pires, 2004*). Along the sampling year 2019, in the autumn–winter period, mean value of air temperature was 11.98 °C, with values ranging between 3.31 °C and 22.42 °C. Mean precipitation was 77.51 mm, with values ranging between 34.5 mm and 145.75 mm, being December the month with more precipitation. In the spring–summer months, mean values

of air temperature were 19.16 °C, ranging between 4.13 °C and 29.70 °C. During these months, the mean precipitation was 31.03 mm, with values ranging between 3.3 mm and 117.2 mm (http://www.ipma.pt/pt/index.html). Monthly values for seawater temperature, salinity and wave height at the sampling area can be found in Fig. 2. Values showed on Fig. 2 were measured every 60 min by an oceanographic buoy (model Seawach) placed at 42.12°N and 9.43°W. Values of temperature and wave height were measured during 2019 but, for salinity there are many lacks of data for 2019 and thus, we included the data of 2018. All the data were obtained from the public repository of Puertos del Estado webpage (http://www.puertos.es/es-es/oceanografia/Paginas/portus.aspx).

## Sampling

The sampling design included two seasons and three sampling dates within each season: spring-summer (April, June and September) and autumn-winter (January, October and December). On each of these dates, the six studied shores were visited, and at each shore, two sites separated by 10 s of meters were randomly selected. At each site, the percentage of mussel's cover was calculated in four quadrats (50 × 50 cm). Cover was estimated by dividing each quadrat into 25 sub-quadrates of 10 ×10 cm, attributing a scale from 0, with an absence of mussels, to 4, when all the quadrat (10 × 10 cm) were covered by mussels, and accumulation up the 25 estimates (*Dethier et al., 1993*). In addition, four measures of mussel clump thickness were randomly taken within each of the four 50 × 50 cm quadrats by introducing a rigid ruler until reaching the bottom of the clump. Within each one of the 50 × 50 cm quadrats, one sample of 10 ×10 cm was collected, by scraping all the mussels in this area, and stored in labelled bags. These samples were preserved in a mixture of formalin 4% with rose of Bengal neutralised with sodium tetraborate anhydrous (Borax). In the laboratory, samples were washed in a tower of sieves of 1,000 µm, 500 µm, 250 µm, 125 µm and 63 µm mesh size. For each sample, twenty random mussels from the 1,000 µm sieve were separated to be measured with a calliper. Measured mussels were grouped in the following size classes: Class 1: <10 mm, Class 2: 10–20 mm, Class 3: 20–30 mm, Class 4: 30–40 mm, Class 5: 40–50 mm and Class 6: >50 mm. Additionally, 10 mussels per replicate were opened with a surgical blade and the shell was separated from the soft body. Afterwards, they were dried and weighed to calculate the condition index as the ratio between soft tissues dry weight and the shell dry weight. Residues retained in the sieves of 500 µm and 1,000 µm were sorted in a dissection microscope and all the mussels were collected and counted. To evaluate mussels in the spat stage (i.e., mussels with size between 500 and 1,000 µm), as a proxy of recruitment, the number of mussels retained in the 500 µm mesh size was counted. The amount of sediment retained at each mesh size (1,000 µm, 500 µm, 250 µm, 125 µm and 63 µm) was evaluated by drying the sieve content at 65 °C for about 12 h and weighting.

## Data analyses

Differences in the total density, spat density, percentage cover, mussel size classes and the amount of sediment retained on mussel clumps (total and different sediment sizes) were examined with analyses of variance (ANOVA). These analyses were based on a four way

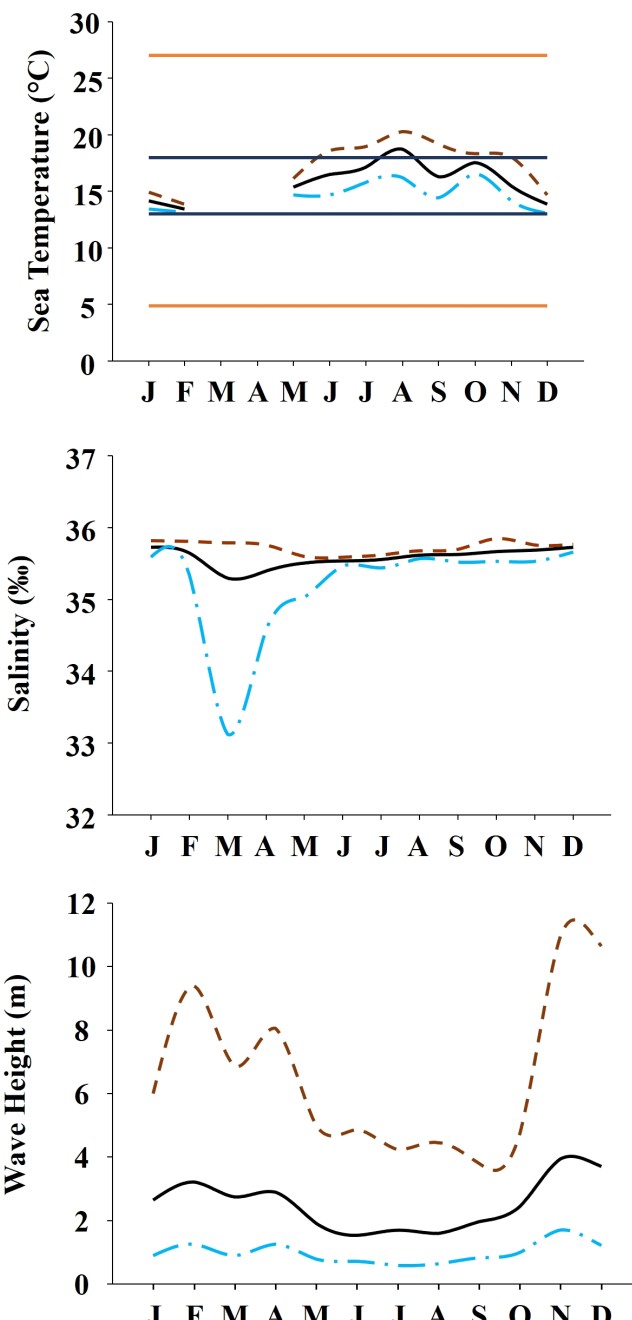

**Figure 2  Environmental data.** Annual variability of environmental variables registered along the year at the studied area. (A) Temperature during 2019. Orange solid lines show the upper and lower limits of temperature tolerance for *M. galloprovincialis*. Blue solid lines show the upper and lower limits of optimal temperature for *M. galloprovincialis*. (B) Salinity during the year 2018. (C) Wave height during the year 2019. In all the figures dashed brown line represents maximum monthly values, black solid line represents average monthly values and dash and dot blue line represents minimum monthly values.

model including Season (Se) with two levels: autumn–winter and spring–summer, as a fixed orthogonal factor, Date (Da) with three levels, as a random factor nested in Se, Shore (Sh) with six levels: Aguda, Valadares, Cabo do Mundo, Moledo, Carreço and Forte do Cão, as a fixed orthogonal factor and Site (Si) with two levels, as a random factor nested in all the previous factors, considering four replicates per site. ANOVA was also used to test for differences on thickness and condition index, but these analyses were based on a five-way model, including all the factors above mentioned and Plot (Pl), as an additional random factor, nested in Se x Da x Sh x Si, with four and 10 replicates for thickness and condition index, respectively. Cochran's C test was done to test for homogeneity of variances. In some cases, when Cochran's C test was significant ($p < 0.05$) data were Ln(X+1) transformed with the objective of removing heterogeneity of variances. When transformation was not possible, untransformed data were analysed and results were considered only if significant at $p < 0.01$, to compensate for the increased probability of type I error (*Underwood, 1997*). Whenever ANOVA demonstrated significant differences ($p < 0.05$) a post hoc Student-Newman-Keuls (SNK) test was done to explore differences. Finally, in order to explore the relationship between total sediment content and mussel descriptors, rank correlation analyses were done for each descriptor. Due to the non-normal distribution of the data, Spearman's rank correlation was used.

# RESULTS

## Spatial and temporal patterns of mussel descriptors

ANOVA results for the percentage cover detected significant differences among Shores (Table 1). Post hoc analysis was not able to provide an alternative hypothesis, probably due to its lower power (*Underwood, 1997*), (Fig. 3). For total and spat density, the interaction between shore and date (Sh x Da) was significant (Table 1). Post hoc analysis for total density showed a homogeneous pattern for the first three dates (Jan, Apr, Jun) but, significant differences were detected in the last three months (Sep, Oct, Dec) (Fig. 4). Results for spat stage showed a similar pattern as total density, but only Aguda showed significant differences in the last two months (Oct, Dec) (Fig. 5).

Condition index and clump thickness showed significant differences for the interaction between shore and date (Sh x Da) (Table 2). Post hoc analysis showed significant differences in the first four months (Jan, Apr, Jun, Sep). On the other hand, in October, Valadares showed significantly higher values than the remaining shores (Fig. 6). For bed thickness, post hoc analysis showed significant differences in the first two months (Jan, Apr) for several shores (Fig. 7).

The size classes 1 and 3 did not show significant differences among seasons or shores (Table 3). However, ANOVA detected significant differences for the interaction between shore and date (Sh x Da) in Classes 2, 4, 5 and 6 (Table 3). The post hoc analyses showed significant differences between shores in June and December for Class 2 (Fig. S1). For Class 4, significant differences were detected in June and October (Fig. S2). For the Class 5, post hoc analysis pointed out differences in January, April, September and December, (Fig. S3). For Class 6, post hoc analysis showed significant differences in January, October. (Fig. S4).
**Table 1** Summary of univariate analyses of variance (ANOVA) for percentage cover, density and spat density of *M. galloprovincialis*.

| Source of variation | df | % Cover | | Density | | Spat density | | F versus |
|---|---|---|---|---|---|---|---|---|
| | | MS | F | MS | F | MS | F | |
| Season=Se | 1 | 1,262.53 | 0.66 | 0.14 | 0.01 | 0.62 | 0.02 | Da(Se) |
| Date(Se)=Da(Se) | 4 | 1,902.84 | **7.14**[***] | 10.07 | **13.56**[***] | 26.04 | **20.39**[***] | Si(SexDaxSh) |
| Shore=Sh | 5 | 1,340.44 | **2.86**[*] | 4.05 | 2.34 | 5.57 | 2.13 | ShxDa(Se) |
| Site(Sh)=Si(SexDaxSh) | 36 | 266.62 | **2.55**[***] | 0.74 | **3.31**[***] | 1.28 | **3.14**[***] | Residual |
| SexSh | 5 | 260.39 | 0.56 | 1.92 | 1.11 | 4.70 | 1.79 | ShxDa(Se) |
| ShxDa(Se) | 20 | 468.49 | 1.76 | 1.73 | **2.33**[*] | 2.62 | **2.05**[*] | Si(SexDaxSh) |
| Residual | 216 | 104.59 | | 0.22 | | 0.41 | | |
| Total | 287 | | | | | | | |
| Transform | | none | | Ln(X + 1) | | Ln(X +1) | | |
| Cochran's test | | $C = 0.0702$ | n.s. | $C = 0.0829$ | n.s. | $C = 0.0820$ | n.s. | |

**Notes.**
Significant differences indicated in bold.
n.s.; not significant.
[*] $p < 0.05$.
[**] $p < 0.01$.
[***] $p < 0.001$.

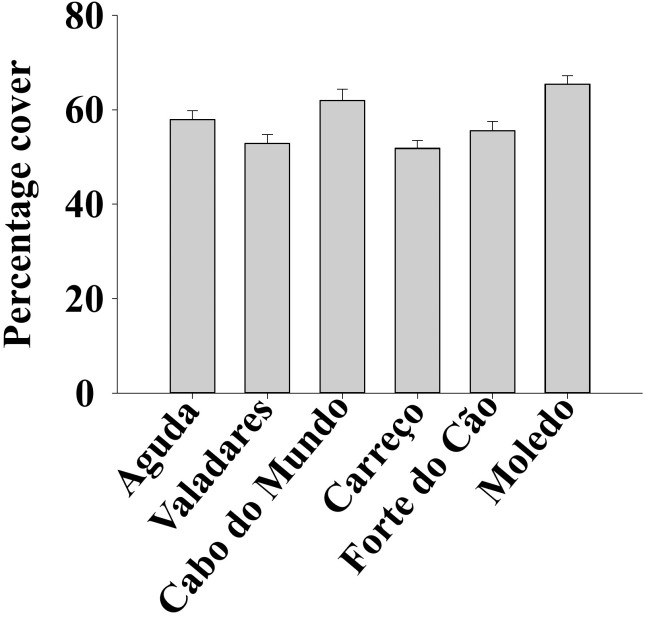

**Figure 3** **Percentage cover of *Mytilus galloprovincialis* at different studied shores.** Mean values (+ SE) of percentage cover in 50 × 50 cm.

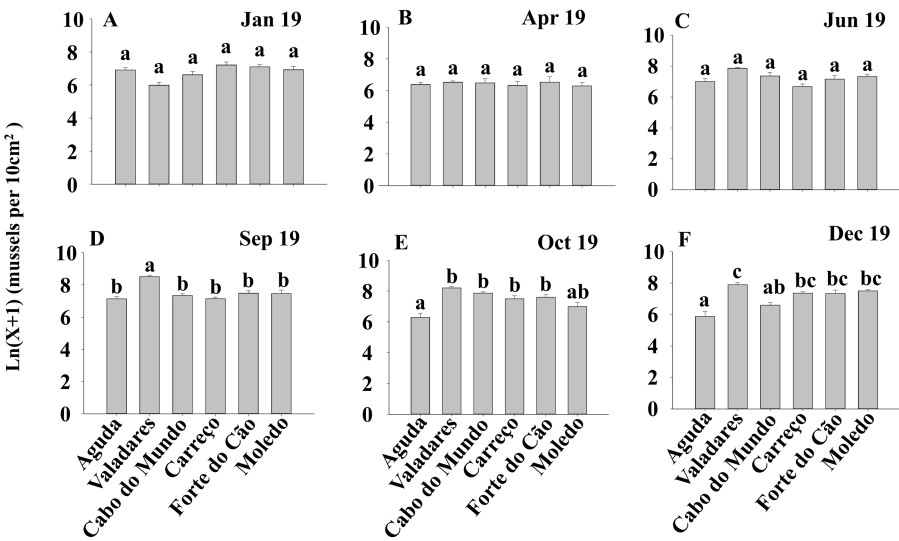

**Figure 4** (A-F) Significant differences of density of *Mytilus galloprovincialis* at different shores and dates. Density of *Mytilus galloprovincialis* at different shores and dates. Mean values (+ SE) of density (number of mussels per 10 cm²). Different lowercase letters indicate significant differences between shores ($P < 0.01$) as detected by SNK test.

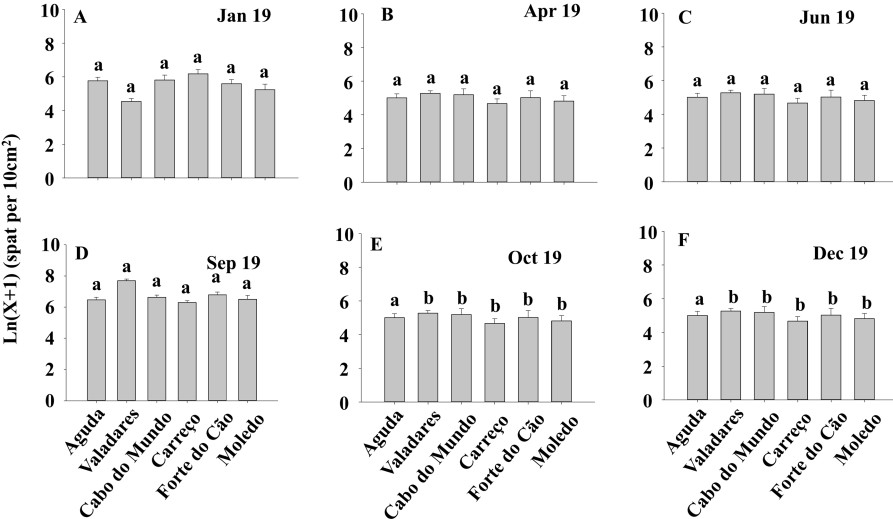

**Figure 5** (A-F) Density of *Mytilus galloprovincialis* in spat stage at different shores and dates. Mean values (+ SE) of mussels in spat stage (number of mussels with size between 500 and 1,000 μm per 10 cm²). Different lowercase letters indicate significant differences between shores ($P < 0.01$) as detected by SNK test.

**Table 2** Summary of univariate analyses of variance (ANOVA) for condition index and clump thickness of *M. galloprovincialis*.

| Source of variance | df | Condition Index | | Clump Thickness | | F versus |
|---|---|---|---|---|---|---|
| | | MS | F | MS | F | |
| Season=Se | 1 | 0.0028 | 0.12 | 0.0413 | 0.03 | Da(Se) |
| Date(Se)=Da(Se) | 4 | 0.0232 | **18.40**[***] | 1.4722 | **6.98**[***] | Si(SexDaxSh) |
| Shore=Sh | 5 | 0.0143 | **2.71**[*] | 2.6067 | **5.47**[**] | ShxDa(Se) |
| Site(SexDaxSh)=Si(SexDaxSh) | 36 | 0.0013 | 0.92 | 0.2109 | **3.55**[***] | Pl(SexDaxShxSi) |
| Plot(SexDaxShxSi)=Pl(SexDaxShxSi) | 216 | 0.0014 | **3.28**[***] | 0.0593 | **1.64**[***] | Residual |
| SexSh | 5 | 0.0011 | 0.20 | 0.2221 | 0.47 | ShxDa(Se) |
| ShxDa(Se) | 20 | 0.0053 | **4.17**[***] | 0.4767 | **2.26**[*] | Si(SexDaxSh) |
| Residual | 259,2 | 0.0004 | | 0.0362 | | |
| Total | 287,9 | | | | | |
| Transformation | None | | | Ln(X+1) | | |
| Cochran's test | | C =0.0744 | s. | C = 0.0398 | n.s. | |

Notes.

Significant differences indicated in bold.

s., significant; n.s., not significant.

[*]$p < 0.05$.

[**]$p < 0.01$.

[***]$p < 0.001$.

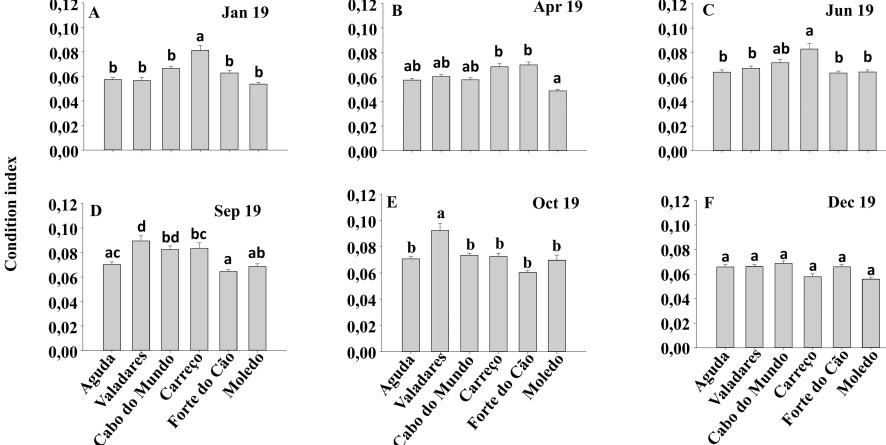

**Figure 6** (A-F) Condition index of *Mytilus galloprovincialis* at different shores and dates. Mean values (+ SE) of condition index. Different lowercase letters indicate significant differences between shores ($P < 0.01$) as detected by SNK test.

The sediment content in clumps differed significantly among shores both for total sediment and different sediment sizes (Table 4) (Fig. 8).

## Relationship between sediment content in clumps and different mussel attributes

Spearman's rank correlations showed that total sediment content significantly increased with mussel clumps thickness ($R = 0.551$, $p < 0.01$). Moreover, a significant positive correlation was found between total sediment content and condition index ($R = 0.136$,

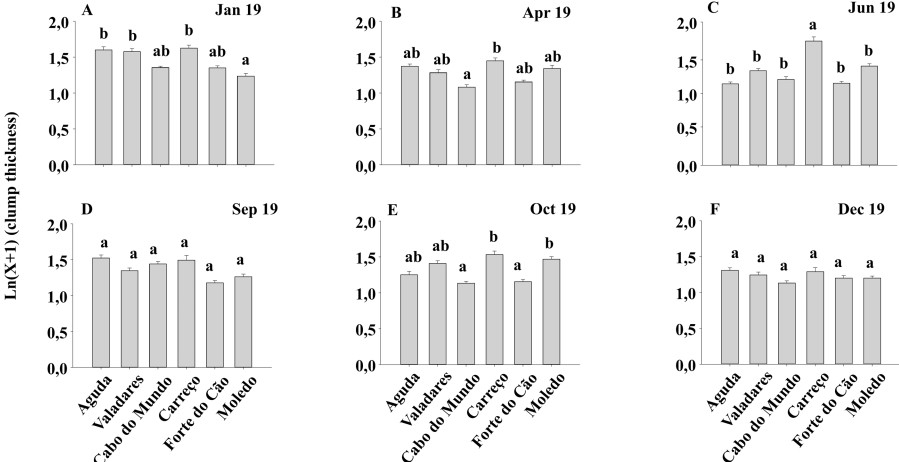

**Figure 7** (A-F) Clump thickness of *Mytilus galloprovincialis* at different shores and dates. Mean values (+ SE) of clump thickness in cm. Different lowercase letters indicate significant differences between shores ($P < 0.01$) as detected by SNK test.

$p < 0.05$) and with the average mussel size ($R = 0.18$, $p < 0.05$) but the correlation coefficients were much lower than that obtained for thickness. However, correlation between total sediment and cover ($R = 0.0356$, $p > 0.05$), density ($R = -0.05$, $p > 0.05$) or spat stage ($R = -0.0345$, $p > 0.05$) were not significant.

## DISCUSSION

This study documented the lack of seasonality for all the studied descriptors in *M. galloprovincialis* along the north coast of Portugal. However, significant differences were shown on the term Sh x Da for density, spat stage, clump thickness, condition index and size classes, demonstrating that drivers shaping these descriptors have limited effects at spatial (i.e., shores) and temporal (i.e., date) scales. Moreover, percentage cover and sediment content on mussel clumps showed significant differences just among shores indicating that the different drivers that could influence these descriptors, have a constant effect along the explored time scales (Season and Date), changing only at spatial scale.

The comparison of our results with previous studies is difficult due to the lack of information about spatial–temporal variability of *M. galloprovincialis* populations in the Iberian Peninsula. However, studies done in other regions (e.g., south–west England and Italian coast) demonstrated differences between seasons for the condition index of *M. galloprovincialis* (*Orban et al., 2002*; *Ivankovic et al., 2005*). Moreover, studies on benthic assemblage seasonality done in north of Portugal were focused on tide pools (*Rubal et al., 2011*; *Bertocci et al., 2012*) and pointed out that benthic assemblages (dominated by macroalgae) showed significant differences between seasons. Results of these studies contrast with our findings for *M. galloprovincialis* suggesting that drivers that shape mussel populations probably are different to those influencing tide pool macroalgae assemblages. In contrast with our results, other invertebrates considered as ecosystem engineers, such as

Ramos-Oliveira et al. (2021), *PeerJ*, DOI 10.7717/peerj.11499

**Table 3 Summary of univariate analyses of variance (ANOVA) for different size classes of *M. galloprovincialis*.**

| Source of variation | df | Class 1 | | Class 2 | | Class 3 | | Class 4 | | Class 5 | | Class 6 | | F versus |
|---|---|---|---|---|---|---|---|---|---|---|---|---|---|---|
| | | MS | F | MS | F | MS | F | MS | F | MS | F | MS | F | |
| Season=Se | 1 | 0.68 | 0.83 | 0.70 | 0.19 | 0.59 | 0.01 | 0.08 | 0.07 | 0.06 | 0.95 | 0.59 | 0.01 | Da(Se) |
| Date(Se)=Da(Se) | 4 | 0.82 | **9.79**[***] | 3.75 | **10.16**[***] | 104.95 | **8.25**[***] | 1.13 | 1.29 | 0.06 | 0.54 | 48.98 | **120.57**[***] | Si(SexDaxSh) |
| Shore=Sh | 5 | 0.08 | 0.52 | 1.16 | 1.25 | 15.02 | 0.68 | 2.98 | 1.82 | 0.70 | 2.24 | 1.44 | 1.40 | ShxDa(Se) |
| Site(Sh)=Si(SexDaxSh) | 36 | 0.08 | 0.95 | 0.37 | **2.49**[***] | 12.72 | **1.65**[*] | 0.87 | **2.33**[***] | 0.11 | 0.85 | 0.41 | **2.45**[***] | Residual |
| SexSh | 5 | 0.18 | 1.18 | 0.60 | 0.64 | 8.06 | 0.36 | 0.80 | 0.49 | 0.23 | 0.73 | 1.44 | 1.40 | ShxDa(Se) |
| ShxDa(Se) | 20 | 0.15 | 1.84 | 0.93 | **2.53**[**] | 22.25 | 1.75 | 1.64 | **1.88**[*] | 0.31 | **2.82**[**] | 1.02 | **2.52**[**] | Si(SexDaxSh) |
| Residual | 216 | 0.09 | | 0.15 | | 7.69 | | 0.37 | | 0.13 | | 0.16 | | |
| Total | 287 | | | | | | | | | | | | | |
| Transformation | | none | | Ln(X+1) | | none | | Ln(X+1) | | Ln(X+1) | | none | | |
| Cochran's Test | | $C = 0.2500$ | s. | $C = 0.0622$ | n.s. | $C = 0.0584$ | n.s. | $C = 0.0636$ | n.s. | $C = 0.0761$ | n.s. | $C = 0.9161$ | s. | |

**Notes.**

Significant differences indicated in bold.

s., significant; n.s., not significant.

[*]$p < 0.05$.

[**]$p < 0.01$.

[***]$p < 0.001$.
**Table 4 Summary of univariate analyses of variance (ANOVA) for sediment content of *M. galloprovincialis* clumps.**

| Source of variation | df | 63 µm | | 125 µm | | 250 µm | | 500 µm | | 1000 µm | | Total | | F versus |
|---|---|---|---|---|---|---|---|---|---|---|---|---|---|---|
| | | MS | F | MS | F | MS | F | MS | F | MS | F | MS | F | |
| Season=Se | 1 | 0.07 | 0.01 | 2.21 | 0.77 | 2.23 | 0.59 | 8.87 | 1.97 | 0.47 | 0.25 | 1.39 | 0.49 | Da(Se) |
| Date(Se)=Da(Se) | 4 | 5.37 | **2.86*** | 2.86 | 2.50 | 3.80 | 1.41 | 4.50 | 2.03 | 1.90 | 1.31 | 2.84 | 1.49 | Si(SexDaxSh) |
| Shore=Sh | 5 | 14.11 | **5.28**** | 5.04 | **2.86*** | 24.21 | **11.54****** | 22.94 | **6.28**** | 11.34 | **5.51**** | 17.78 | **7.42****** | ShxDa(Se) |
| Site(Sh)=Si(SexDaxSh) | 36 | 1.88 | **2.22****** | 1.14 | **3.13****** | 1.99 | **3.05****** | 2.22 | **2.82****** | 1.45 | **3.80****** | 1.90 | **3.68****** | Residual |
| SexSh | 5 | 2.67 | 1.00 | 1.05 | 0.60 | 2.21 | 1.06 | 1.30 | 0.36 | 1.65 | 0.80 | 1.64 | 0.68 | ShxDa(Se) |
| ShxDa(Se) | 20 | 2.67 | 1.42 | 1.76 | 1.54 | 2.10 | 1.05 | 3.66 | 1.65 | 2.06 | 1.41 | 2.40 | 1.26 | Si(SexDaxSh) |
| Residual | 216 | 0.84 | | 0.36 | | 0.65 | | 0.79 | | 0.38 | | 0.52 | | |
| Total | 287 | | | | | | | | | | | | | |
| Transformation | | none | | Ln(X+1) | | Ln(X+1) | | Ln(X+1) | | Ln(X+1) | | Ln(X+1) | | |
| Cochran's Test | | $C = 0.2446$ | s. | $C = 0.0789$ | n.s. | $C = 0.0564$ | n.s. | $C = 0.0691$ | n.s. | $C = 0.0521$ | n.s. | $C = 0.0590$ | n.s. | |

**Notes.**

Significant differences indicated in bold.

s., significant; n.s., not significant.

*$p < 0.05$.

**$p < 0.01$.

***$p < 0.001$.

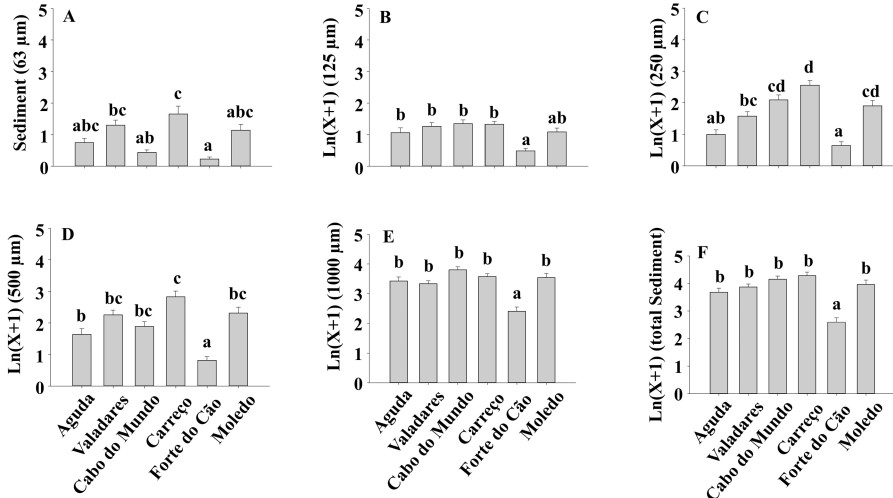

**Figure 8 (A-F) Sediment content of *Mytilus galloprovincialis* clumps at different shores.** Mean values (+ SE) of sediment content in grams. Different lowercase letters indicate significant differences between shores ($P < 0.01$) as detected by SNK test.

*Sabellaria* worms (*Gravina et al., 2018*), showed seasonality with a growing phase beginning in summer until fall, a degeneration phase in winter and spawning in spring (*Peharda et al., 2007*; *Gravina et al., 2018*).

*Mytilus galloprovincialis* showed a homogenous pattern along time for percentage cover. The absence of seasonality or other temporal variability in this descriptor suggests that mussels occupy similar areas of rocky shores along the year, independently of the season and date. The study by *Boaventura et al. (2002)*, done along the Portuguese coast, showed

that the abundance of *M. galloprovincialis* declined from north to south of Portugal; this is in agreement with the high values of the percentage cover found in our study. The percentage cover is determined by the abundance and size of individuals, which is affected by growth, mortality and recruitment (*Roughgarden, Iwasa & Baxter, 1985*; *Petraitis, 1995*). Therefore, results for the percentage cover are in concordance with our results for density and spat stage that did not show any seasonality. The constant amount of spat stage along the year suggests a constant recruitment and spawning for *M. galloprovincialis* in the studied area. There are many studies about the reproductive cycle of *M. galloprovincialis* showing contrasting conclusions (e.g., *Seed, 1976*; *Nagarajan, Lea & Goss-Custard, 2006*). *Seed (1976)* suggests that spawning in *Mytilus* is regulated by a combination of external and internal factors but, the variation of seawater temperature is the most crucial one and that low temperatures reached in winter and autumn can inhibit mussel spawning. *Seed (1976)* pointed out that optimum seawater temperature for the genus *Mytilus* ranges between 10 °C and 20 °C, which can influence not just the spawning but also the growth. For the specie *M. galloprovincialis* the optimal temperature for the population growth rate ranges between 13 °C and 18 °C (*Yoann & Cédric, 2018*). As temperature is different between regions and may change considerably from year to year (*Seed, 1976*), the reproductive cycle of *M. galloprovincialis* can present different seasonal patterns according to the studied region and even among years. On the other hand, the study by *Ceccherelli & Rossi (1984)* does not exclude the probability of spat stage occurs independently at different times along the whole year, with different values among months. Therefore, *M. galloprovincialis* can present different reproductive phenology at different regions (*Ceccherelli & Rossi, 1984*). Constant recruitment and reproduction along the year in our studied area is possible since seawater temperature is around the optimum range described by *Yoann & Cédric (2018)* during all year. The few significant differences detected for total and spat abundance could be the result of local natural or anthropogenic disturbances.

The condition index of mussels is affected through a variety of intrinsic and extrinsic factors, like food availability, sea temperature, salinity and gametogenic cycle (*Okumus & Stirling, 1998*). Many studies (e.g., *Çelik et al., 2012*) state that the condition index shows a seasonal variability; achieving a maximum during gonadal development and decreases when spawning period starts. However, other studies (e.g., *Orban et al., 2002*) showed that the condition index registered a similar value along the year, in concordance with our results. The lack seasonal variability suggest that mussels from north of Portugal do not have a defined period for gonadal development and spawning. Therefore, mussels seem to be able to spawn along the whole year, explaining in this way the constant recruitment that suggests the spat density results. As percentage cover, clump thickness provides information about the habitat furnished by *M. galloprovincialis* for the other species. The percentage cover gives information about the habitat provided by *M. galloprovincialis* along the shore (horizontal) while thickness depends on the accumulation of different layers of mussels, providing three-dimensional habitat (vertical). The clump thickness of mussel populations in the intertidal is strongly influenced by wave action (*Zardi, McQuaid & Nicastro, 2007*). Therefore, the magnitude of the hydrodynamic forces along the year can affect the thickness of mussel clumps due to dislodgment of thick clumps during storms

(*Zardi et al., 2006*). Results showed that mussel clumps thickness differs among shores and dates but not between seasons. Therefore, seasonal changes on wave height seems to play a minor role in shaping the clump thickness since the wave height, in the studied area, has a clear seasonal pattern (i.e., higher during Winter–Autumn) in contrast with our findings for clump thickness. Therefore, observed differences among dates and shores may be the result of local shoreline characteristics.

The size classes also showed absence lack of seasonality. The growth of mussels can be assessed by the size, which usually is related to age, but it is also dependent on environmental conditions (*Seed, 1976*). *Peharda et al. (2007)* showed that the growth of *M. galloprovincialis* is slow at temperatures above 21 °C and at temperatures below 10 °C, but also found that food availability can be an important driver for the growth of mussels. The lack of seasonal pattern on the abundance of different size classes can be the result of a continuous recruitment and growth due to optimal environmental conditions on the studied area. In general the lack of seasonal changes on the *M. galloprovincialis* population on the studied area contrast with the seasonal changes on water and air temperature, wave action or upwelling events related with the considered seasons. However, the recorded changes on temperature in the north of Portugal are around the optimum range of *M. galloprovincialis*. Moreover, *Abrantes & Moita (1999)* showed significant differences in the phytoplankton biomass, the main food source for *M. galloprovincialis*, between August and January along the coast of Portugal. Despite a reduction on the phytoplankton biomass during winter, this reduction is not so marked as in the centre or south of Portugal (*Abrantes & Moita, 1999*). Therefore, the availability of food during the winter in the north of Portugal may explain the high percentage cover in the north observed by *Boaventura et al. (2002)* and the lack of seasonal variability observed in our study.

However, significant differences were detected among shores or among shores and dates. These differences could be caused by non-seasonal physical and biological processes such as trampling, harvesting, pollution events, predation, or parasitism that could vary between dates and shores independently of the season. For example, the effects of harvesting and trampling could also vary among shores, for instance, in function of their accessibility to humans (*Smith & Murray, 2005*; *Rius, Kaehler & McQuaid, 2006*; *McPhee, 2017*). However, differences in the morphology or the orientation to wave exposure of the studied shores could be among plausible drivers for the descriptors of percentage cover and sediment content that only showed significant differences between shores.

Results of our study also showed a lack of variability on sediment accumulation along the two studied temporal scales (i.e., season and date), although significant differences were found between shores. Similar results were found in Argentina (*Dos Santos et al., 2018*) where no seasonal pattern in the sediment accumulated on mussel beds was found. However, the amount of sediment that reaches the shores can vary seasonally, as found by previous studies on South Africa (*Zardi et al., 2006*) and Argentina (*Dos Santos et al., 2018*). Therefore, despite the variable supply of sediments, results of our study and that by *Dos Santos et al. (2018)* suggest a limited and constant capacity of mussel beds to retain sediment under natural conditions. However, the ability to accumulate sediment showed significant differences among rocky shores in our study. These results contrast with those found

by *Dos Santos et al. (2018)* where no significant differences were found on the sediment accumulated on mussel beds between their studied sites. In our study we have considered a broader spatial scale than that in the study by *Dos Santos et al. (2018)*. Therefore, at each shore, local currents, bottom topography, tidal speed and different sources of sediment may have resulted in the heterogeneous spatial pattern found. Moreover, our results showed a positive relationship between sediment accumulation and mean mussel size, condition index and clump thickness, however, the correlation coefficients of average mussel size and condition index were very low and only clump thickness, with a correlation coefficient above 0.5, may play a relevant role in shaping sediment accumulation. Results also showed that 500 $\mu$m and 1,000 $\mu$m sediment size categories were the most abundant ones. This observation, is in agreement with the sediment size found in some northern Portuguese sandy beaches (*Veiga et al., 2014*) suggesting that sandy beaches and rocky shores receive sediment from the same or similar sources. It is also noticeable the low amount of 63 $\mu$m sediment accumulated because burial by fine sediments induces higher mortality than burial by coarse sediments (*Hutchison et al., 2016*). Moreover, *Zardi et al. (2006)* showed that *M. galloprovincialis* has higher tolerance to burial than *Perna perna*, probably because its larger palpial labs are more efficient sorting particles and avoiding gill damage.

## CONCLUSIONS

The lack of seasonality and reduced temporal variability found for *M. galloprovincialis* in the study area may be due to the optimal water temperature and food availability along the whole year. The lack of temporal variability on the amount of sediment accumulated on mussel clumps suggest constant input of sediment that only differs among shores and may be influenced by clump thickness.

## ACKNOWLEDGEMENTS

We are grateful to the editor Mark Costello, Sandra Fiori and an anonymous referee for helpful comments and suggestions, which greatly improved this paper. We are also very grateful to Dr. Laura Guerrero for her help with figures.

### Funding
This research was developed under the Project No. 30181 (PTDC/CTA-AMB/30181/2017), co-financed by COMPETE 2020, Portugal 2020 and the European Union through the ERDF, and by FCT-Foundation for Science and Technology through national funds within the scope of UIDB/04423/2020 and UIDP/04423/2020. The funders had no role in study design, data collection and analysis, decision to publish, or preparation of the manuscript.

### Grant Disclosures
The following grant information was disclosed by the authors:
ERDF.
FCT-Foundation for Science and Technology: UIDB/04423/2020, UIDP/04423/2020.

## Competing Interests

The authors declare there are no competing interests.

## Author Contributions

- Catarina Ramos-Oliveira performed the experiments, analyzed the data, prepared figures and/or tables, authored or reviewed drafts of the paper, and approved the final draft.
- Leandro Sampaio performed the experiments, authored or reviewed drafts of the paper, and approved the final draft.
- Marcos Rubal conceived and designed the experiments, performed the experiments, analyzed the data, prepared figures and/or tables, authored or reviewed drafts of the paper, and approved the final draft.
- Puri Veiga conceived and designed the experiments, analyzed the data, prepared figures and/or tables, authored or reviewed drafts of the paper, and approved the final draft.

## Data Availability

Raw measurements are available in the Supplementary File.

## Supplemental Information

Supplemental information for this article can be found online at http://dx.doi.org/10.7717/peerj.11499#supplemental-information.

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
