# Peer review of "Spatial-temporal variability of Mytilus galloprovincialis Lamarck 1819 populations and their accumulated sediment in northern Portugal"

_PeerJ, doi:10.7717/peerj.11499_

## Round 0.1 · original submission · Major Revisions

I have received too good and consistent reviews. The MS needs a lot of work, care and improved presentation and writing. Thus I recommend Major Revision and additional review.

Reviewer 1 ·

Basic reporting

The English needs major revisions by a mother tongue. I made some comments below but there are many more examples and some sentences are hard to follow, making the manuscript sound in part repetitive and redundant, and in part confusing the reader that has to go over the same sentences many times before grasping the meaning.

The introduction is generally too long and while it provides a wide range of references (although none very new) and ideas , it seems largely unfocused jumping from the importance to some various threats and only living a little space to the study itself. I wonder if it could be more focused to the key questions and gaps.
The discussion can also be more structured
The conclusion is also unfocused, with repetitions of results that should not be in a conclusion section.

Experimental design

This is a well designed monitoring study on population dynamics of Mytilus galloprovincialis in Portugal.
Unfortunately it lacks essential complementary environmental metrics - especially with regards to hydrodynamics, which makes the study weaker as not much can be explained and thus the discussion is full of speculative sentences.
It is a shame because it is very interesting that there are not many seasonal effects, especially with regards to spat or loss of cover from wave action (although i do not understand whether the shores are very exposed or locally sheltered).

Validity of the findings

The study could be better introduced and discussed in a wider context. It is lacking of a strong focal discussion point.
More environmental data to relate these findings could be useful and an additional assets.
In alternative I suggest to revisit some aspects related to explaining the findings, since there are no data: for example the byssal strength in relation to waves cannot be explained simply by the thickness of the clumps. So is the optimal temperature theory that is not tested and it comes out as a strong result according to the conclusions.

Additional comments

abstract
L22 Marine repeated two times
L27 descriptors is vague here
L 28 what does along the year mean? Better to esplicitate the cadence of sampling
L 30 results showed
L33 lack of seasonality for what?

introduction
L 44 I would refrain from using subjective words such as ‘important’
L 45 they can control - it is too ‘active’ and they don’t control it on purpose
L 53 probably as a result?
l 56 I am not sure that they have such a high value. they are much less costly than other bivalves-
L 56 it is vulnerable to rather than exposed
L 59 see major comment about English

L 78 what is it meant by high water circulation? Strong flow rates / wave action?
L 79-83 English

methods
l170 if i understand correctly winter was split in two ‘winters’ (beginning of the year and end of the year) - this should be better rationalised in the methods

discussion
here also the english should be revised throughout
the reference in L 324 should read Ceccherelli

L 360 - were the clumps studied in exposed parts of the shore? and byssal strength was not measured. It is not necessary that thickness increases but they may produce more threads

L 387 - can none of this be tested?

L 408 is there no model of wave action locally or the possibility of adding dynamometers on the shore?


Figures: are the letters to indicate significant differences between sites also there to indicare significant differences between months ?



Raw data - JaAg12: sample in January, in Aguda, shore 1, site 1 I think this should be site 2? If understand from the other examples

Reviewer 2 ·

Basic reporting

The Results section is quite lengthy for reading. Repetitive by saying that "this statistically significant", when a more integrative description if preferred. Small statistical differences are not biologically relevant, and authors should only emphasize those of interest for their work.

The Discussion section is very long, considering obtained results. Authors should try to integrate the results obtained for each endopoint and not describing them individually.

Experimental design

The authors separate samples in two season blocks: autumn-winter and spring-summer. This makes that January 2019 is grouped with October and December 2019. That is, months separarted almost for one year are grouped. Authors do not provide information regarding environmental variables at each month, that allow to group them together. Only a mention to two previous works is done in Lines 150-153. Environmental data collected at each month should be also included as Supplementary Table or in the Raw data file.

In the methods section a Table should be included describing the characteristics of each sampling site. Sediment type (rocky, sandy, muddy..), existing impacts (industries, urban, agricultural, protected área..). This allows a better knowledge of the environmental conditions existing at each site.

Validity of the findings

The absence of significant seasonal or site differences regarding biological endpoints is very odd. Authors mention that previous works in the same area described seasonal changes in mussel physiology. Considering that mussels in the North West Iberian Peninsula spawn during spring and summer time, it is not clear why the mussels analyzed in the present study showed a constant condition index.

It is well known that mussels inhabiting environments rich in nutrients can produce gametes all along the year. Nutrient sources could be natural (constant phytoplankton production) and/or artificial (organic materials arriving from rivers or urban pollution). Author mention that changes in phytoplankton content were previously described in the study area (Lines 159-160 and 379-383), but overall they assume that nutrient status is optimal all along the year in their mussel populations. What about organic material pollution sources? Could they contribute to the constant condition indesx determined in studied mussel populations? are the CI values determined in those mussels high, medium or low in comparison to data existing in the literature?

Additional comments

The manuscript is well written and clear. The organization of the contents is also correct. The experimental approach is correctly described. A more detailed information regarding sampling points should be included in the Study Area section.
Some of the results, biological endpoints mainly, need some more details. For example the absence of changes in CI requires a better interpretation and find an association with existing environmental variables.

---

## Round 0.2 · Major Revisions

Thank you for the improvements to the text. I ask you to more critically revise the paper based on the referees' constructive and helpful suggestions. For example, please include the environmental data to provide the context for the study as suggested by both referees. Surely the seasonal temperature is critical to the biology for example? Also, please include a good quality copy showing the study area in the main paper. This is key to place it in a geographic context. Just because it is not analysed as a correlate is not a reason not to provide it. Several times you answer referees' comments in the rebuttal (response). This cannot help the reader. Seeing as how the referees raise these questions indicates you should clarify these issues in the paper.

---

## Round 0.3 · Minor Revisions

Both referees are positive about the revised paper and one makes some good minor suggestions for improvements. I emphasise that now is time to make the figures ready to publish. The colours on Fig 2 are not ideal (consider colour blind readers) and dashed, dotted lines could also be used. The text on figures 5 to 8 is minimal and would be better to double its size because the figures will probably be reduced in the journal.

Reviewer 1 ·

Basic reporting

The article has improved, it is now easier to read and follow through. I appreciate the authors effort.

Experimental design

The authors have made the corrections suggested, including adding environmental data.

Validity of the findings

The addition of the conclusion is appreciated, despite this is 'speculative' it is interesting for future research.

·

Basic reporting

This interesting paper tries to fill knowledge gaps of spatial-temporal variability of Mytilus galloprovincialis populations along the Atlantic coast of the Iberian Peninsula and the accumulated sediments on their clumps.

The title clearly reflects the contents and the abstract is sufficiently informative.

I am not a native English-speaking person, but I think the language and grammar are clear.

The literature cited is well referenced and relevant.

Introduction
I have included some minor corrections in the attached PDF.

Lines 65-71 - I suggest expanding this paragraph where the effect of environmental factors on mussel populations is mentioned since this is the main objective of the paper. Also, provide more information about the interest of evaluating the accumulation of sediments on these coasts, that is, if it is because a natural accretion-erosion process affects them or if it is due to increases in turbidity due to anthropic activities

Figures
Figure 1- I suggest improving the quality of this figure and include the map coordinates.
Figure 2 – I suggest change the label for this sentence: Annual variability of environmental variables registered along the year at the studied area. (A) Temperature during 2019. Red and green lines show the range of tolerance and optimal temperature of Mytilus galloprovincialis respectively. (B) Salinity during the year 2018. (C) Wave height during 2019.
Figure 3- Missing space between words Mean// values and of//percentage
Figure 4, 5, 6 and 7 – I suggest bring the y axes of all the figures to the same maximum value.
Figure 8 - Missing space between words Mean// values. Check the title of the y-axis of figure 8A.

Experimental design

In the description of the study areas, it is necessary to clarify why these beaches were selected as different factors for the analysis. It is if the chosen beaches are under different anthropic pressures (harvesting, trampling, etc.), or have some particular environmental characteristic. Sampling and Data analysis are clear and sufficient to support the research.

Validity of the findings

Discussion

Line 280- Please expand this sentence, indicate if there are differences in anthropic pressure between beaches. What indicators allow you to reach this type of conclusion?

Lines 298-300 -I do not agree with this explanation. For this study, wave measurements were not taken at each beach, there is only a general wave record from a buoy. Although this wave record indicates that wave height varies seasonally, the intensity of the waves at each beach is likely to be different due to local shoreline characteristics.

---

## Round 0.4 · Minor Revisions

Thank you for the revisions. However, the figures remain sub-optimal. Fig 2 with text too small on X-axis (just use large J F M A M etc.), reduce clutter by giving salinity in whole numbers (33, 34, 35, 36). The legend for what the lines are does not need to be repeated 3 times and not spelling mistakes (thermal). This is really basic figure production and should have been done before the first submission. Fig 3 and 4 could be rotated sideways and place names given in full (thereby avoiding abbreviations). Figures 4 to 8 if better arranged can reduce clutter by not repeating labels on axes. In Abstract this sentence can be omitted because it is not really "crucial" ["Therefore, it is crucial to investigate its spatial and temporal patterns of variability because there is a lack of information about this species along the Iberian Peninsula."] The related Introduction text is good thank you.

---

## Round 0.5 · accepted · Accept

Thank you for the improved figures and abstract.